# Risk Assessments and Risk Mitigation to Prevent the Introduction of African Swine Fever into the Danish Pig Population

**DOI:** 10.3390/ani14172491

**Published:** 2024-08-27

**Authors:** Jan Dahl

**Affiliations:** Danish Agriculture & Food Council, DK-1609 Copenhagen, Denmark; jd@lf.dk; Tel.: +45-24635877

**Keywords:** African swine fever, truck wash, risk mitigation, wild boar

## Abstract

**Simple Summary:**

The Danish pig industry, in collaboration with the authorities, has implemented several risk-reducing measures to prevent the introduction of African Swine Fever into Danish pig production. Ongoing risk assessments are conducted to adjust or establish new measures. The first outbreaks in a new area in Northern Europe often occur in wild boar, likely due to discarded food waste at rest areas and in forested areas. A fence has been established on the Danish–German border to limit the migration of wild boar from Germany to Denmark, and the limited population of wild boar in Denmark has been culled. All trucks that transport pigs or cattle are cleaned and disinfected at the border if they have been abroad, according to the DANISH Transportstandard. Through continuous contact with the feed industry, changes in the importation of feedstuffs and other auxiliaries are analyzed, and risk assessments for each feed item are conducted. Based on this, according to the Danish Product Standard, it is not permitted to store or use feed items such as hay, straw, silage, or bedding imported from areas with outbreaks of African Swine Fever on Danish properties with pigs. These types of feed items can contain parts of dead animals in which the virus can survive for a long time. Oil cakes, fats, and oils imported from countries with African Swine Fever may only be used after heat treatment.

**Abstract:**

African Swine Fever poses a significant threat to pig production. An outbreak in Denmark would have severe economic consequences, potentially resulting in a loss between 335 million and 670 million euros. To mitigate the major risk factors, the Danish authorities and the Danish pig industry have implemented several risk-reducing measures. The small wild boar population in Danish nature has been culled, and a fence between Denmark and Germany has been constructed to stop or reduce the risk of migrating wild boar from Germany. All trucks arriving from abroad, intended for the transport of Danish pigs, are inspected, washed, and disinfected at facilities near the border before being allowed to transport pigs between herds or from herds out of Denmark. Cross-border trade of feedstuffs and potentially risky materials is continuously monitored. Based on risk assessments, feed types or bedding materials from countries with African Swine Fever that could contain parts of wild boar are banned from Danish pig herds. Certain types of fats and oils from countries with African Swine Fever can only be used after heat treatment. The import of whole kernels of corn, wheat, or barley is not considered a risk.

## 1. Introduction

In June 2007, African Swine Fever (ASF) was confirmed in the Caucasus region of Georgia after many years of being confined to Africa and the Italian island of Sardinia. It gradually spread north and west from the Caucasus, primarily through wild boar, and due to the transport of infected pigs, or feeding pigs or wild boar contaminated pork products. Since 2014, ASF has been circulating in wild boar populations in the Baltic countries and Poland, and it has been introduced into several Eastern European countries, reaching as far west as Belgium, Italy, the Czech Republic, and Germany, and as far north as Sweden.

Table 1 summarizes the number of outbreaks in Europe from 2022 to August 2024, based on official statistics from the EU [1].

Danish pig production is export-based, with an annual production of between 28 and 32 million pigs over the last 10 years [2]. Between 30% and 50% of the pigs produced are exported as weaners or growers to many European countries, and 50% to 70% are finished and slaughtered in Denmark. With an annual export of more than 1 million tons of pork, Denmark would be severely affected by an outbreak that hinders the exportation of pigs and pork.

The Department of Food and Resource Economics, Copenhagen University [3], estimated that an outbreak of African Swine Fever would cause an economic loss for Danish pig production of EUR 335 million if the outbreak is in a pig herd, and EUR 670 million if wild boar become infected. An outbreak in wild boar would affect exportation to third countries for more than a year [4]. 

This makes it vital for the Danish pig industry to keep African Swine Fever out of Denmark. To achieve this, formal and informal risk assessments are conducted continuously to mitigate the risks of introduction, considering the changing epidemiological picture and changes in the trade patterns of potential fomites, like feed and other materials, crossing borders between countries.

The Danish Veterinary and Food Administration can mitigate some of these risks, but some risks can only be handled by the industry. This does not imply a conflict between the industry and the authorities; rather, there are different responsibilities and opportunities for risk mitigation for both parties.

The Danish pig industry has developed the Danish Product Standard and the Danish Transport Standard [5], which cover more than 95% of production. An important part of these standards is the implementation and regulation of several biosecurity measures. One-third of all herds are visited each year by a third-party auditor to ensure compliance.

## 2. Risk Factors

The World Organisation of Animal Health (WOAH) has identified several pathways for the introduction of African Swine Fever into pig herds or wild boar populations, including direct contact between infected and non-infected pigs, feeding garbage containing infected pork, as well as via vehicles, clothes, and implements [6]. Contaminated feed is another possible pathway, as shown by Dee [7]. The Danish pig industry and the Danish veterinary authorities have taken several steps to mitigate these risks, including mandatory washing and disinfection of returning livestock trucks, eliminating free-living wild boar from Danish territory, establishing a fence at the Danish–German border, and establishing industry rules for the handling of feed items that might constitute a risk.

In Northern Europe, wild boar has been a major factor in the spread of African Swine Fever. The first cases in many Northern European countries occurred in wild boar (Belgium, Germany, Sweden, and the Czech Republic) [8]. In several European countries, the infection has persisted in wild boar for years [9], demonstrating that successful eradication of African Swine Fever in wild boar populations is difficult once the infection has become widespread. However, eradication was achieved in some countries when the outbreak had a limited geographical extent (the Czech Republic and Belgium) [10]. Soft ticks (Ornithodoros) play an important role in the epidemiology of African Swine Fever in warmer climates, but are not part of the epidemiology in Northern Europe [11]. This subject is not discussed further herein.

## 3. Wild Boar in Denmark

Wild boar became extinct in Denmark at the beginning of the 19th century due to a combination of hunting, cold winters, and lack of suitable habitats [11]. However, since 2008, wild boars have been observed in low numbers in Denmark [11].

In 2018, the Danish Ministry of Food, Agriculture and Fisheries and the Danish parliament decided, in close collaboration with the Danish pig industry, that the small population of wild boar should be removed, and a fence should be built at the border between Denmark and Germany to prevent the migration of wild boar from Germany to Denmark. The Danish Nature Agency estimated the total population of wild boar in Denmark as 150–200 [12].

## 4. The Wild Boar Fence between Denmark and Germany

In 2019, the construction of the 68 km-long fence was completed, with 20 openings for traffic and numerous openings that can be passed by smaller animals and humans (ladders, small holes, and cattle grids) (Figure 1). In 2021, the last wild boar in the free range was shot; in total, 157 were either trapped or shot between 2018 and 2021. Since then, there have been fewer than a handful of sightings of wild boar; some were shot, and others likely migrated south again.

The cost of the fence was approximately EUR 6 million, according to the Danish Nature Agency [12]. 

## 5. Mitigating the Risk of Introduction through Importing or Exporting Pigs

Denmark has a limited importation of live pigs, averaging less than 200 pigs annually, mostly boars or gilts for breeding. These are subject to normal importation procedures according to EU regulations [13]. Additionally, the Danish pig industry has implemented a voluntary quarantine and testing program, with high compliance so far.

In 2023, Denmark exported more than 15,000,000 pigs, mainly to Poland and Germany, including areas where African Swine Fever was present. More than 25,000 trucks were used for these exports, posing a risk when returning to Denmark to pick up pigs again. EU regulations [14] demand that all means of transport used for animal transport must be cleaned and disinfected after unloading. However, controlling the efficiency of cleaning and disinfection is challenging.

After the foot-and-mouth outbreak in the UK in 2001, Danish authorities required returning trucks to undergo additional inspection, cleaning, and disinfection at authorized washing facilities after crossing the Danish border [15,16]. Ten years later, the EU commission intervened, claiming the extra control, cleaning, and disinfection violated EU regulations on free trade.

Based on experiences in the washing facilities, Danish pig and cattle organizations decided that this control, cleaning, and disinfection were necessary to ensure an adequate level of hygiene. Thus, the pig industry formed the DANISH Transportstandard [5] as an adjunct to the DANISH Produktstandard to replace the regulation from Danish authorities.

All herds delivering pigs to major Danish slaughterhouses are part of the Danish Product Standard, which is compliant with the German QS scheme (Quality Scheme for Food). This means pigs exported to Germany are also covered by the Product Standard and the Transport Standard. More than 95% of Danish production is covered by the Danish Product Standard.

Trucks returning to Denmark must go through one of three authorized facilities at the border. Each truck is inspected on arrival, and if no potentially contaminated material is found inside, the truck is washed on the outside and disinfected inside and out. GPS data from the truck is uploaded and analyzed. Based on the GPS information, a certificate is issued electronically and on paper.

The Danish Transport Standard divides Europe into three zones: infected, buffer, and free zones. If GPS data show a truck has been in an infected zone, the certificate indicates a 7-day quarantine after leaving the washing facility before it can load pigs from a Danish farm. If a truck has been in a buffer zone, it has a 48 h quarantine. If it has only been in free zones, it can pick up pigs for export directly from a herd.

Trucks with 48 h or 7-day quarantines can always pick up pigs from a collection point, where pigs are unloaded from a truck only used for domestic transport and then loaded onto a truck transporting pigs across the border. If an export truck is used for domestic transport, it always has a 48 h quarantine, even if it has only been in a free zone. If it has been in an infected zone, the quarantine is 7 days.

All movements of live pigs in Denmark must be registered in the official Danish pig movement database. The Danish Agriculture & Food Council cross-references this database with certificates issued at washing facilities at the border to ensure that quarantine requirements have been fulfilled.

The cost of extra control, washing, and disinfection for Danish pig production is estimated to be approximately EUR 3 million. 

## 6. Mitigating the Feed Risk for the Introduction of African Swine Fever

Swill feeding with infected, non-heat-treated pork products is likely the main pathway for the introduction of African Swine Fever into new areas. Although not clearly proven, it is probable that this was the main introductory pathway into the Dominican Republic, several countries in Eastern and Central Europe, and Southeast Asia.

EU regulation 1069/2009 [17] prohibits the use of food and kitchen waste or swill for feeding pigs and other farm animals. In large parts of Northern and Western Europe, there have been no introductions into farmed pigs by this route. However, unintentional introduction through some feed matrices could occur. African Swine Fever can survive for long periods in pig tissues [6]. Hay, silage, wrapped grass, and similar feed types can be contaminated by tissues of dead pigs or wild boar during harvest. Using these feed types could infect pig herds if introduced into the herd. To mitigate this risk, the use, or even presence, of these feed types is not allowed on pig farms, according to the Danish Product Standard, if the feed is imported from areas with African Swine Fever.

Dee [7] showed that the African Swine Fever virus can remain infectious under conditions simulating realistic transport conditions for trans-Atlantic and trans-Pacific passage in, for example, soybean meal and soy oil cake. The Danish pig industry closely monitors changing trade patterns of feed ingredients in collaboration with feed mills. Whenever changes in trade patterns are suspected or revealed, representatives from the feed mills cooperation (DAKOFO) and representatives from the pig production industry meet and discuss the circumstances.

Denmark is normally self-sufficient in grain production, but in 2018, a drought reduced Danish grain production, necessitating the importation of 1 to 1.5 million tons of wheat or corn, potentially from countries with African Swine Fever in both wild boar and domestic pigs.

A literature search in 2018 did not reveal any studies on the risk of introducing African Swine Fever to pig herds using whole corn or wheat for feed production. Heat treatment to 56 °C for 70 min or 60 °C for 20 min can mitigate the risk, but this was not a realistic option [6].

Due to the lack of experimental data, historical trade patterns of wheat and barley for feed production from areas with African Swine Fever were investigated. Large volumes of grain for feed production had been exported from Russia, Ukraine, and Eastern Europe to many parts of the EU without causing any outbreaks. Transport times from these areas of Europe to Denmark were at least 14–18 days [18].

Furthermore, outbreak data from the EU database showed that most outbreaks in pig herds in Europe occurred from June to September [19]. If newly harvested grain contributed to outbreaks, it would have been for a short period after harvest. Based on these epidemiological findings and the transport time to Denmark, it was concluded that the risk of introducing African Swine Fever through the importation of wheat or corn was limited.

In 2024, the EFSA published a study examining different feed matrices. Live viruses could only be isolated for a short period from wheat, barley, and grass, supporting the epidemiological findings [20].

In 2018 and 2019, African Swine Fever was spreading in several East Asian countries. Denmark imports organic soya and several oils from these countries. As Dee [7] showed, some of these matrices could be contaminated and contain infectious viruses for several weeks to months. Based on these results, it was decided that soya and certain fats and oils from these regions could only be used after heat treatment. Fats and oils that are solid during transport and require heating for unloading do not need further heat treatment. These requirements were implemented in the Danish Product Standard [5].

## 7. Discussion and Conclusions

Mitigating the risk of introducing porcine pathogens into a country with a substantial pig production and a large export of live pigs is challenging. The Danish authorities and the Danish pig industry have worked closely together to mitigate these risks. Denmark has eliminated wild boar from Danish nature areas, built a fence at the Danish border, implemented industry rules for empty livestock trucks coming back to Denmark after unloading pigs abroad, and implemented risk-mitigating measures for the importation of feed and bedding material. Compared to the estimated cost of an outbreak of EUR 335–670 million, the cost of these mitigating measures is relatively small.

## Figures and Tables

**Figure 1 animals-14-02491-f001:**
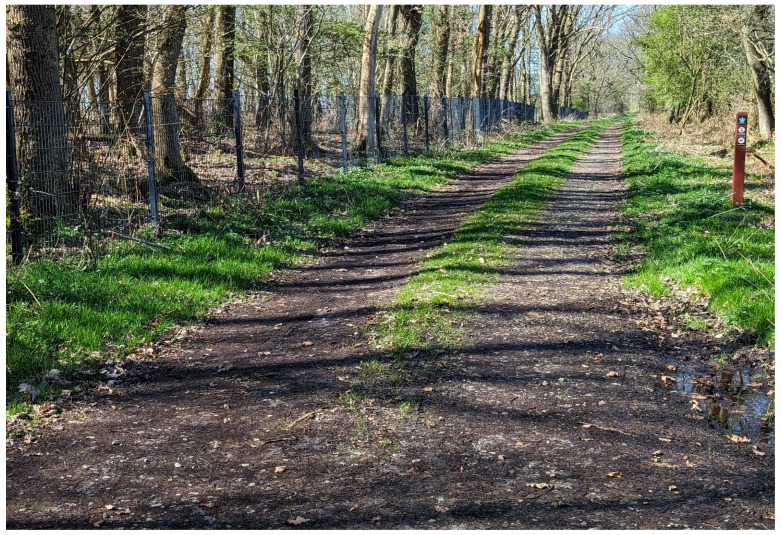
The wild boar fence between Denmark and Germany. The fence is 1.5 m high and has been dug 0.5 m into the ground (photo courtesy of Dr. Gordon Spronk).

**Table 1 animals-14-02491-t001:** Registered outbreaks in domestic pigs and wild boar in Europe.

Country	2024 1 January–7 August		2023		2022	
	Domestic Pigs	Wild Boar	Domestic Pigs	Wild Boar	Domestic Pigs	Wild Boar
Estonia		12	2	53		57
Latvia	4	516	8	730	6	913
Lithuania	6	376	3	439	16	302
Poland	35	1165	30	2744	14	2152
Italy	10	1106	17	1047	4	277
Ukraine	34	11	37	9	7	2
Czech Republic		26		56		1
Romania	122	108	740	292	329	465
Hungary		245		407		550
Bulgaria	1	95	3	322	2	387
Slovakia	1	97		546	5	550
Serbia	116	82	991	213	107	146
Moldova	8	6	19	6	14	3
Montenegro		1				
Bosnia-Hercegovina	27	36	1508	22		
Croatia	3	38	1124	11		
North Macedonia	2	36	15	41	30	9
Kosovo			9	4		
Belgium						
Germany	9	279	1	899	3	1628
Sweden		8		60		
Greece	5	17	6	2		
Albania		2				
Total	383	4262	4513	7903	537	7442

## Data Availability

Only official statistics mentioned in the reference sections was used.

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
