# Peer review of "Risk Assessments and Risk Mitigation to Prevent the Introduction of African Swine Fever into the Danish Pig Population"

_animals, 2024, doi:10.3390/ani14172491_

Round 1

Reviewer 1 Report

Comments and Suggestions for Authors

Please see the attached document for comments and suggestions for improvement.

Author Response

Thank you for your comments. It was very useful. Just to clarify: The paper is the result of a request from Dr.s Dee and Spronk for the special issue of Animals. They wanted a paper describing the initiatives taken by the Danish pig industry and the Danish authorities. 

Reviewer comments:

  • Key words have not been included on the title page

Done

  • Line 57 – Please include World Organisation of Animal Health prior to referring to it by it’s acronym WOAH (in the first instance).

Done

  • Line 57 – add clarification: are the pathways for the introduction of African swine fever only into the farmed pig population, or is this generalised to include the introduction of ASF into all suidae (inc. wild boar)?

Done

  • Line 61 – can you add a summary of the “several steps” that have been put in place by the Danish veterinary authorities to mitigate the risks?

Done

  • Line 95 – Please add reference to the EU regulations, or include a written summary of the import procedures/regulations.

Done

  • Line 107 – is this comment supported by any publications or communications?

I have included two orders, one from when the official regulations were implemented in 2001 and the order, that cancelled the official regulation in 2010. I am not aware of any public documents from EU on the subject, but we were informed by the Veterinary & Food Administration, that it was cancelled because of communication from the EU commission.

  • Line 111 – Why is DANISH capitalised, is this an acronym?

No, it is not an acronym. But it is the way it is spelled in the official documents from the Danish pig production.

  • Line 113 – “Danish” not capitalised.

Not sure, what you mean? Here Danish refers to the Danish authorities. I think, the correct way of spelling that is Danish

  • Lines 138-139 – is there a publication that supports this comment?

To the best of my knowledge, there is no definitive proof, that this was the way of introduction. As there is no definitive proof, that this was the way of introduction into wild boar in Sweden, Belgium. Obviously, this is very difficult to proof. But there is anecdotal evidence, that ASF-virus has been found at border-control in several countries.

  • Line 170-171 – Reference the EU database. Additional comment on the timeframe for infection: As most outbreaks appear to occur between June and September, could this be included in some way in risk management and disease mitigation?

Sorry, I forgot to include the EFSA-reference. Done. From a Danish perspective, I would say, that was what we did, when we imported grain from Ukraine in 2018. Apart from that, I think it could definitely be used in countries, where it has become endemic. I feel, that it would

  • Line 175 – include reference to EFSA study.

Done

  • It would be nice to include a concluding statement at the end of the report that summarises the key factors involved in mitigating and preventing ASF introduction.

Done

  • Question to authors: Have you considered any management of viral vectors (i.e. ticks?) to limit and mitigate potential for spread?

Ticks are not a problem in Northern Europe.  So, I am including this information in the risk-factor section.

Reviewer 2 Report

Comments and Suggestions for Authors

African Swine Fever spread from the Caucasus region in 2007 to various parts of Europe, posing a significant threat to pig populations and industries. The manuscript titled "Risk Assessments and Risk Mitigation for the Prevention of Introduction of African Swine Fever into the Danish Pig Population" discusses various measures implemented by the Danish pig industry to prevent the introduction of ASF. The study provides a description of various risk mitigation measures, including biosecurity standards, wild boar management, and feed safety protocols. The objective of preventing ASF in the Danish pig population is clearly stated and aligns with the measures discussed. The study design is fundamentally sound but would benefit from additional quantitative data and detailed implementation descriptions. More detailed information on the implementation and monitoring of these measures would be helpful.

1.     The manuscript could benefit from more quantitative data to support the effectiveness of the measures, such as statistics on ASF outbreaks in neighboring countries, risk assessments, and the impact of mitigation measures.

2.     The manuscript could expand on potential future challenges and directions for further strengthening ASF prevention, such as the development of new biosecurity technologies or the impact of global trade changes on ASF risk.

3.     Add an economic perspective to discuss the costs and benefits of the implemented biosecurity measures.

4.    The manuscript offers a novel perspective on ASF prevention in the context of the Danish pig industry. However, more emphasis on innovative techniques or recent advancements in ASF prevention would strengthen the paper.

5.     Tables could be used to summarize and present data in a structured format, making it easier to compare and analyze the information. Each table should be referenced in the text, and the data in the tables should match the descriptions and numbers in the results section. Ensure that all claims are directly supported by the data presented.

Author Response

(The authors gave the same response as above.)

Round 2

Reviewer 2 Report

Comments and Suggestions for Authors

The review paper has been revised, but the content was fewer, it better to improve the detail of the manuscript .